# CAN PAST EXPERIENCE HELP LLMS REASON FASTER?

## ABSTRACT

Allocating more compute to large language models (LLMs) reasoning has generally been demonstrated to improve their effectiveness, but also results in increased inference time. In contrast, humans can perform tasks faster and better with increased experience and exposure. Hence, this paper aims to investigate the question: Can LLMs also become faster at reasoning through recurrent exposure on relevant tasks, and if so, how can it be achieved? To address these questions, we first formalize the problem setting of LLM reasoning speedup systematically in the dimensions of task similarity and compute budget calculation. We then propose SPEEDUPLLM, a theoretically guaranteed framework to implement and benchmark such reasoning speedup behaviour based on adaptive compute allocation and memory mechanisms. We further conduct comprehensive experiments to benchmark such behaviour across different reasoning tasks, question similarity levels, memory methods, and reasoning methods. Results show that LLMs can generally reason faster with past experience, achieving up to a 56% reduction in compute cost when equipped with appropriate memory and reasoning methods.

## 1 INTRODUCTION

Large Language Models (LLMs) have demonstrated reasoning capabilities to solve problems through step-by-step logical thinking (Brown et al., 2020; Wei et al., 2022), which is crucial for applying LLMs to complex tasks in fields such as math reasoning (Achiam et al., 2023). Recently, research shows LLMs can better solve complex problems when allocated more compute at test time (Snell et al., 2025), and such techniques are referred to as *test-time scaling* (Snell et al., 2025; Liu et al., 2025). However, increased compute also brings substantial computational overhead and increased reasoning time (Sui et al., 2025), and this stimulates existing research on efficient reasoning algorithms (Sun et al., 2024; Wang et al., 2025; Ding et al., 2025), efficiency-oriented model fine-tuning (Luo et al., 2025; Yu et al.; Kang et al., 2025; Munkhbat et al., 2025), model compression and distillation (Sun et al., 2025; Zhang et al., 2025a).

For humans, repeated exposure on a specific task can lead to a significant reduction in the cognitive effort and time for execution (Shiffrin & Schneider, 1977; Logan, 1988), which is fundamental for humans to become proficient and efficient in various activities, from reading, motor skills, to complex problem-solving (Anderson, 1982). In cognitive terms, humans often start with slow, effortful reasoning on new problems, but as they repeatedly solve similar problems, this reasoning gradually becomes faster. Memory of past experience is the key enabler of this transition (Langan-Fox et al., 2002). This analogy suggests a concrete question for LLMs: can memory mechanisms help convert compute-heavy, System 2-style test-time scaling into cheaper, more direct, System 1-style reasoning when the model encounters familiar or similar questions? To fill this gap, this paper focuses on the question: **Can LLM reasoning be faster after past experience, and how can this be achieved?** In the remaining of this paper, we refer to LLMs' such potential behaviour as *reasoning speedup*.

There are two main reasons limiting LLM systems from achieving reasoning speedup: 1) **Independent Question Processing**: Regular LLM systems simply process each query independently. However, nowadays, public LLM services accept millions of questions every hour, many of which are related or near-duplicate (Dammu & Alonso, 2024), which are not fully leveraged to reduce redundant compute or accumulate useful experience. 2) **Static Compute Budget Allocation**: Existing test-time scaling methods do not adaptively allocate compute based on an LLM's proficiency to a question, thus hindering the model from becoming faster when meeting familiar questions. For example, Best-of-N

sampling uses a fixed hyperparameter $N$, while tree-search methods, such as Tree-of-Thought, rely on a predefined maximum number of nodes to expand (Yao et al., 2023).

Therefore, to explore whether LLM can achieve reasoning speedup and how to achieve it, we propose SpeedupLLM, a unified framework to formulate, implement, and benchmark the behaviour of "reasoning speedup over experiences" across various LLM reasoning settings. Our core idea is to endow test-time scaling with memory mechanisms, so that over a sequence of similar questions the model can gradually shift from heavy, exploratory reasoning to lighter, System 1-style reasoning that reuses past solutions. Specifically, we first systematically formulate this question as a problem to explore the decreasing trend of reasoning cost across different question similarity levels and reasoning paradigms. SpeedupLLM implements LLM reasoning speedup based on two key elements: 1) Adaptive Compute Budget Allocation, which extends various existing test-time scaling methods by early stopping with a threshold. 2) Memory Mechanism, which appends memory of previous questions and answers after processing each question. We conduct a theoretical analysis to prove that SpeedupLLM can enable reasoning speedup.

We further conduct comprehensive experiments to benchmark different memory methods, test-time scaling methods on achieving LLM reasoning speedup at varying question similarity levels. Experiments show that the LLM reasoning speedup behaviour generally exists across different reasoning tasks, memory and reasoning methods. For similar questions, the reasoning compute budget can be **reduced by up to 56%** with the help of memory mechanisms.

The contribution of this work includes:

- **New Problem**. We identify and systematically formulate the problem of reasoning speedup as the exploration of decreasing compute budget trends across varying question similarity levels, different reasoning and memory methods.
- **Unified Framework**. We propose a unified and theoretically guaranteed framework, SpeedupLLM, to implement and benchmark LLM reasoning speedup based on memory mechanisms and adaptive compute allocation strategies, and it generally supports various reasoning methods.
- **Extensive Experiments**. We conduct benchmarking experiments across four reasoning tasks, four test-time scaling methods, five memory methods, and four levels of question similarity.
- **Findings and Insights**. Our findings demonstrate that LLMs can achieve faster reasoning after experience. Such behaviour generally exists across different settings, and the reasoning compute budget can be reduced by up to 56%.

## 2 RELATED WORK

### 2.1 TEST-TIME SCALING OF LLMS

**Test-Time Scaling** is the technique to improve LLMs' reasoning ability on complex questions by allocating more compute at the test time (Snell et al., 2025), and it has received increasing attention from the research community (Parashar et al., 2025; Wu et al., 2025; Ji et al., 2025; Li et al., 2025; Liu et al., 2025). Current representative test-time scaling methods include 1) parallel scaling methods, e.g., Best-of-N sampling (Stiennon et al., 2020), which samples multiple complete answers and selects the highest-scoring one as the final output, and Self Consistency (Wang et al., 2022), which generates multiple answers and select the most common one; 2) Sequential Scaling methods, e.g. Self-Refine (Madaan et al., 2023; Gou et al., 2023), which gradually refine the answer based on internal or external feedback; 3) Tree Search methods (Yao et al., 2023; Feng et al., 2023; Guan et al., 2025), which usually form each reasoning step as a node, and conduct tree search algorithms to search for optimal reasoning chains; 4) Long Chain-of-Thought methods, e.g., the reasoning of OpenAI GPT-4o (Jaech et al., 2024) and DeepSeek-R1 (Guo et al., 2025), which conduct implicit searching in the text space by generating long reasoning chains.

### 2.2 LLM MEMORY

**Memory mechanisms** enable LLMs to retain and use information to generate responses (Zhang et al., 2024c), and such information can be from past experience or an external knowledge base (Zeng et al., 2024). There are two main forms of memory: parametric form and textual form (Zhang et al., 2024c). Parametric-form memory stores the memory in model weights, with representative methods

including supervised fine-tuning (SFT) (Hu et al., 2022; Shao et al., 2023), which finetunes the LLMs with past inputs and outputs; and knowledge editing (De Cao et al., 2021; Mitchell et al., 2021; Fang et al., 2024), which mainly focuses on injecting factual knowledge. Textual-form memory saves textual information as memory; the content can be original past interactions (Li et al., 2023; Huang et al., 2023b; Liu et al., 2023; Zhong et al., 2024; Zheng et al., 2023b), reflection (insights extracted from past interactions) (Shinn et al., 2023; Renze & Guven, 2024; Yang et al., 2023b; Hui & Tu, 2024), and atomic facts (Anokhin et al., 2024; Li et al., 2024). In this work, we focus on leveraging past experience to enhance efficiency, so we consider past experience-based memory, including SFT, textual-form past experiences, and reflection on past experiences. We exclude fact-oriented memory structures (e.g., atomic fact databases), as they are primarily designed for knowledge recall rather than past experience.

## 3 METHODOLOGY

### 3.1 PROBLEM FORMULATION

In this work, our central question is whether it is possible that, an LLM can gradually become faster when answering a sequence of similar questions. Formally, let $f$ be an LLM, and let $\mathcal{Q} = \left[q^{(1)}, q^{(2)}, \ldots, q^{(N)}\right]$ be a sequence of $N$ test questions, where each $q^{(n)}$ is a natural language question. We study how the similarity among questions in $\mathcal{Q}$ affects the model's reasoning efficiency.

To this end, we define levels of *similarity* of a group of questions from most similar to least similar, as shown in Table 1 in math reasoning tasks as an example.

Table 1: Definition of question similarity levels in math reasoning, from most similar (**S1**) to least similar (**S4**). Definitions in other reasoning tasks are given in Appendix B.

| Level | Description | Example |
|-------|-------------|---------|
| S1 | Exactly the same questions. | "Solve for $x$: $x^2 - 5x + 6 = 0$" vs "Solve for $x$: $x^2 - 5x + 6 = 0$" |
| S2 | Same numbers, different wording | "Solve for $x$: $x^2 - 5x + 6 = 0$" vs "Find the roots of the quadratic equation $x^2 - 5x + 6 = 0$" |
| S3 | Same structure, different numbers | "Solve for $x$: $x^2 - 5x + 6 = 0$" vs "Solve for $x$: $x^2 - 7x + 12 = 0$" |
| S4 | Same underlying knowledge, different structure and numbers | "Solve for $x$: $x^2 - 5x + 6 = 0$" vs "A rectangle has an area of 12 and a perimeter of 14. What are its side lengths?" |

To measure the reasoning efficiency, we define *compute budget*, $\texttt{cost}(f(q^{(n)}))$ for answering $q^{(n)}$, as the number of conducted operations in each test-time scaling method's dominant scaling dimension, e.g., the number of sampled answers in Best-of-N, and the number of nodes expanded in tree search-based methods. In practice, a "non-increasing compute budget" over a sequence $\mathcal{Q}$ simply means that the model does not need to sample *more* answers, expand *more* nodes, or take *more* refinement steps for later questions than for earlier ones, while maintaining similar accuracy.

Our primary objective is to investigate the trend of compute budgets $\left[\texttt{cost}(f(q^{(1)})), \texttt{cost}(f(q^{(2)})), ..., \texttt{cost}(f(q^{(N)}))\right]$ given the questions $\mathcal{Q}$, and identify conditions under which there can be a decreasing trend, considering varying levels of question similarity, different test-time scaling strategies, and memory mechanisms.

### 3.2 SPEEDUPLLM: A UNIFIED FRAMEWORK FOR IMPLEMENTING AND BENCHMARKING LLM REASONING SPEEDUP

To implement and benchmark LLM reasoning speedup, we propose SPEEDUPLLM, a theoretically guaranteed framework that can give a decreasing trend on reasoning cost for relevant questions, based on adaptive compute budget allocation and memory mechanism. Before introducing the formal notation, we first describe the intuition. SpeedupLLM treats each test-time scaling method as a procedure that generates a *sequence* of candidate solutions and scores them. Memory summarizes what worked well on previous questions. When a new question arrives, the model can often recognize

it as similar to past ones, quickly generate a good candidate, and then *stop early* instead of exhaustively exploring all candidates.

### 3.2.1 FRAMEWORK DESIGN

At a high level, SpeedupLLM has two building blocks. First, we view any test-time scaling method as a procedure that generates a *sequence* of candidate answers and scores them; adaptive compute allocation then corresponds to stopping this procedure early once a sufficiently good candidate appears. Second, we add a memory mechanism so that, after solving earlier questions, the model tends to generate high-quality candidates *earlier* in the sequence for later, similar questions. The combination of these two components is what enables reasoning speedup. Crucially, the two components play fundamentally different and complementary roles: memory improves the likelihood that a good candidate appears early, while adaptive allocation converts such earlier good candidates into actual compute savings. Neither component alone can yield reasoning speedup.

**Preliminary.** We first give a general formulation of existing test-time scaling methods. Let $\mathcal{S}$ be the set of *test-time scaling methods*. When the model process the question $q^{(t)} \in \mathcal{Q}$ using a test-time scaling method $s \in \mathcal{S}$, it generates multiple candidate answers in the reasoning process as $\mathcal{R}_s^{(t)} = f_s(q^{(t)})$, where $f_s$ means generating using test-time scaling method $s$ with model $f$, $\mathcal{R}_s^{(t)} = \{r_{1;s}^{(t)}, r_{2;s}^{(t)}, \ldots, r_{n_t;s}^{(t)}\}$, and each $r_{k;s}^{(t)} \in \mathcal{R}_s^{(t)}$ is a candidate answer. An evaluation function $\texttt{score}(\cdot)$ estimates the quality of each candidate, and the final answer is selected via $\arg\max_{r \in \mathcal{R}_s^{(t)}} \texttt{score}(r)$.

In Appendix D, we show how various existing test-time scaling methods can be unified into this form.

**Adaptive Compute Budget Allocation.** To extend existing test-time scaling methods to adaptively allocate compute budget based on the model's proficiency, we aim at strategies to early-stop the generation once a satisfying answer has been generated. This allows a smaller candidate set to achieve the same maximum score as the full set. Equivalently, we want to truncate the candidate sequence as soon as we see a candidate whose score is "good enough", instead of always running the full procedure. Formally, we use $s'$ to denote the adaptive extension of method $s$. Then $f_{s'}$ is attained by:

$$\min_{f_{s'}} \texttt{cost}(\mathcal{R}_{s'}^{(t)}) \quad \text{s.t.} \max_{r \in \mathcal{R}_{s'}^{(t)}} (\texttt{score}(r)) \geq \tau, \tag{1}$$

$$\text{where} \quad \mathcal{R}_{s'}^{(t)} := f_{s'}(q^{(t)}) \tag{2}$$

$$\text{and} \quad \forall r \in \mathcal{R}_{s'}^{(t)}, \ \forall r' \preceq r \ \Rightarrow \ r' \in \mathcal{R}_{s'}^{(t)} \tag{3}$$

where $\tau$ is the threshold that a score is considered as satisfying, and $\preceq$ denotes the generation order in $\mathcal{R}_s^{(t)} = f_s(q^{(t)})$, where $r_1 \preceq r_2$ means $r_1$ is generated no later than $r_2$ in $\mathcal{R}_s^{(t)}$. In Section 3.3, we show how various existing test-time scaling methods can be extended to adaptively allocate compute budget under this formulation.

**Reasoning with Memory Mechanism.** However, adaptive allocation alone does not produce speedup unless the model becomes more confident on later questions; this confidence gain must come from an external source, memory. Next, we incorporate the memory mechanism into the reasoning process. Let $\mathcal{M}$ be the set of *memory methods*. During inference, the model processes each question $q^{(t)} \in \mathcal{Q}$ sequentially, leveraging the current memory state $\mathbf{M}^{(t)}$ as

$$\mathcal{R}_{s';m}^{(t)} = f_{s'}(q^{(t)}; \mathbf{M}^{(t)}), \tag{4}$$

where $m \in \mathcal{M}$ is the employed memory method. The maintained memory $\mathbf{M}^{(t)}$ is constructed from prior question-answer pairs by

$$\mathbf{M}^{(t)} = g_m(\{(q^{(i)}, \mathcal{R}_{s';m}^{(i)})\}_{i=1}^{t-1}) \quad \forall t > 1. \tag{5}$$

where $g_m$ is the function to save memory under the used memory method $m$. Conceptually, $g_m$ decides *what* to remember from past questions, and $\mathbf{M}^{(t)}$ provides additional context that helps $f_{s'}$ generate high-scoring candidates earlier in the sequence for future, similar questions. When this happens reliably, adaptive allocation can stop earlier and save compute on these later questions.

### 3.2.2 THEORETICAL ANALYSIS

Here, we prove that SPEEDUPLLM can reduce reasoning cost as the model experiences more relevant questions. We begin by showing that, under adaptive compute budget allocation, if the answer quality improves over time, then the required compute budget decreases accordingly, as shown in Theorem 1.

**Theorem 1** (Non-Increasing Compute Budget with Non-Decreasing Answer Quality). *Let $s$ be a test-time scaling method producing a sequence of candidates $\mathcal{R}_s^{(t)} = \{r_1, r_2, \dots\}$. Let $s'$ be the adaptive extension. Let $\mathcal{R}_{s'}^{(t)}$ denote the actual subset generated by $s'$. Assume:*

1. *The cost function $\mathrm{cost}(\mathcal{R})$ is increasing with respect to the size $|\mathcal{R}|$.*

2. *The probability that the best response exceeds the quality threshold, i.e. $\mathbb{P}\big(\max_{r \in \mathcal{R}_s^{(t)}} \mathrm{score}(r) \geq \tau\big)$, is non-decreasing with $t$.*

*Then, the expected compute budget of the adaptive method $s'$ is non-increasing with $t$:*

$$\mathbb{E}[\mathrm{cost}(\mathcal{R}_{s'}^{(t+1)})] \leq \mathbb{E}[\mathrm{cost}(\mathcal{R}_{s'}^{(t)})].$$

In words, if the model becomes more likely to produce a high-scoring candidate somewhere in the sequence as $t$ grows, then an early-stopping version of the same test-time scaling method will, on average, need no more and typically fewer samples/search steps for later questions.

*Proof.* Elaborated in Appendix C.1. $\square$

Next, we show that the memory mechanism can help improve answer quality.

**Theorem 2** (Non-Decreasing Answer Quality with Accumulating Relevant Memory). *Assume the memory mechanism provides first-order stochastic dominance improvement on individual candidate scores. That is, for any candidate index $j$ in the generation order of $s$:*

$$\mathbb{P}(\mathrm{score}(r_{j;s}^{(t+1)}) \geq x) \geq \mathbb{P}(\mathrm{score}(r_{j;s}^{(t)}) \geq x) \quad \forall x.$$

*Further, assume the candidates in $s$ are generated under one of the following structural conditions:*

**(C1)** **Independence:** *Candidates are generated independently.*

**(C2)** **Monotonic DAG:** *Candidates follow a DAG structure where a child's score distribution stochastically improves if its parents' distributions improve.*

*then for any $k$, the probability of a satisfying answer appears in the first $k$ candidates, i.e., $\mathbb{P}\left(\max_{1 \leq i \leq k} \mathrm{score}(r_{i;s}^{(t)}) \geq \tau\right)$, is non-decreasing by $t$.*

Intuitively, Theorem 2 formalizes a mild assumption: as the memory accumulates relevant experience, it should not make any candidate worse and should tend to make some candidates better, so the chance that a good candidate appears among the first $k$ positions increases over time.

*Proof.* Elaborated in Appendix C.2. $\square$

With Theorem 1 and Theorem 2, we have

**Corollary 1.** *Under the assumptions of Theorem 2, the adaptive method $s'$ defined in Theorem 1 achieves a non-increasing expected compute budget.*

*Proof.* Elaborated in Appendix C.3.

This corollary shows that, by integrating adaptive compute allocation and memory-augmented reasoning, SPEEDUPLLM can yield a decreasing trend in the compute budget of reasoning. Put together, the results say: if memory helps the model find good candidates earlier (Theorem 2), then an early-stopping version of test-time scaling will automatically spend no more and often less compute on later questions (Theorem 1), which is exactly the notion of reasoning speedup we aim to study. Intuitively, the main assumption of the theoretical grarantee, which is that the stored memory from earlier questions does not hurt the performance in later query questions, may hold when queries are highly similar in answers, but may not hold if superficially similar questions have divergent answers or if memory induces overfitting.

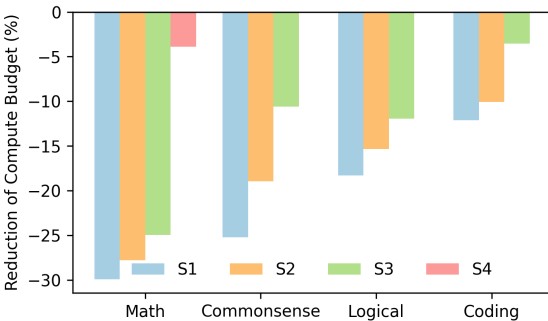

Figure 1: Compute budget changes by question similarity levels (S1 for most similar, S3 or S4 for least similar) on four types of reasoning tasks, averaged over memory methods and reasoning methods.

### 3.3 Specification to Different Test-Time Scaling Methods

Next, we show how SpeedupLLM can be specified for different test-time scaling strategies. We explore four representative streams of test-time scaling methods: Best-of-N, Tree-of-Thoughts (Yao et al., 2023), Self-Refine (Madaan et al., 2023), and Long Chain-of-Thought (Long CoT) (Chen et al., 2025; Guo et al., 2025). These modifications make each scaling strategy adaptive, allowing them to allocate compute budget based on the LLM's familiarity with the questions.

**Best-of-N**. In this method, the minimal unit of generation after which a complete answer can be evaluated is a whole answer. Thus, $\text{cost}(\cdot)$ is measured by the number of generated and evaluated answers. $\text{score}(\cdot)$ is provided by an LLM Judge (Zheng et al., 2023a) or a Process Reward Model (Lightman et al., 2023; Zhang et al., 2025b). Optimizing Eq. 3 is practically performed by sequentially (or in batches) generating and scoring each answer. The generation stops once an acceptable answer is produced.

**Tree-of-Thoughts**. In this method, the minimal unit of generation after which a complete answer can be evaluated is a node in the search tree (often representing a reasoning step). Thus, $\text{cost}(\cdot)$ is evaluated as the number of generated and evaluated nodes. Similar to Best-of-N, $\text{score}(\cdot)$ is given by an LLM Judge (Zheng et al., 2023a) or a Process Reward Model (Lightman et al., 2023; Zhang et al., 2025b). To practically optimize Eq. 3, when expanding each node in the tree search, we sequentially evaluate each node; once we encounter a node with an above-threshold score, we prune the following nodes and expand the current node. Note that this approach also unifies DFS and BFS.

**Self-Refine**. This method is intrinsically compute adaptive. In this method, the minimal unit is one whole refined answer. Thus, $\text{cost}(\cdot)$ is evaluated as the number of generated and evaluated answers. $\text{score}(\cdot)$ is given by the model itself or an external LLM. Since this method automatically stops when a satisfying answer appears, there is no specific modification required to optimize Eq. 3.

**Long CoT**. Long CoT reasoning conducts free-form reasoning processes that implicitly incorporate self-refinement and tree-search strategies within the text generation space (Chen et al., 2025). This approach allows the model to continue generating content until it determines that a satisfactory answer has been reached, eliminating the need for predefined stopping tokens such as "wait" (Sui et al., 2025); thus, it is also an inherently adaptive method. Since this is a text-space reasoning method, the $\text{cost}(\cdot)$ is evaluated as the total number of generated tokens, and $\text{score}(\cdot)$ is assessed by the model's own estimation of the next-token probability, i.e., to generate another "wait" to continue thinking or stop with the current answer. This method also does not require specific modification to be compute budget-adaptive.

## 4 Experimental Setup

**Benchmarking Dimensions.** To explore the LLM reasoning speedup behaviour, we conduct experiments along four dimensions: 1) Task Domain, 2) Question Similarity, 3) Memory Method, and 4) Scaling Method. The details are introduced as follows.

**Data.** In this study, we create a dataset covering four task domains. We use MATH (Hendrycks et al., 2021) (math reasoning task), HumanEval (Chen et al., 2021) (coding task), CommonsenseQA (Talmor et al., 2018) (commonsense reasoning task) and ProntoQA (Pan et al., 2023) (logical reasoning task)

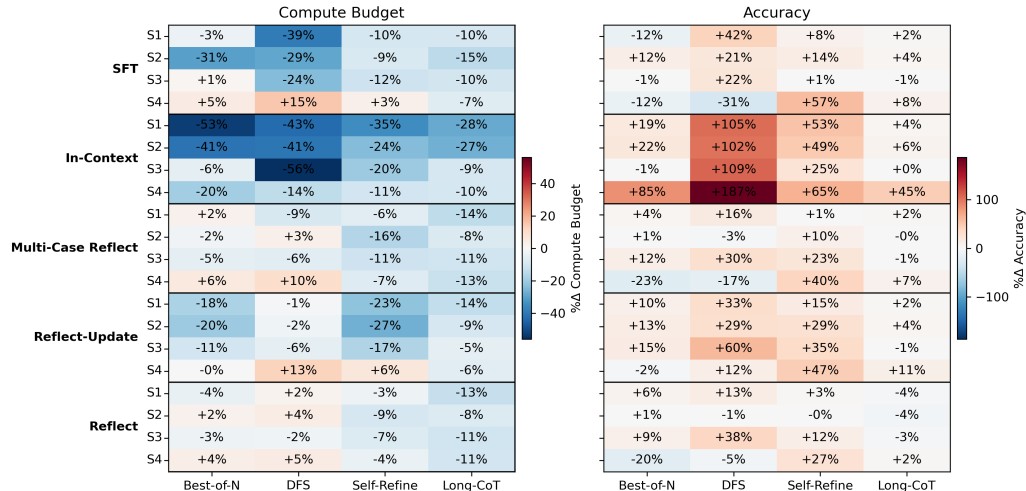

Figure 2: Changes in compute budget and accuracy relative to the baseline without memory mechanisms, across each memory method, test-time scaling method, and question similarity level in the math dataset. Values represent the relative compute budget and accuracy, averaged over all question backbones and variations. Results show that efficiency improvement is general across different settings, and the improvements of efficiency and accuracy are highly correlated.

datasets as data sources. We randomly sample 10 questions from the MATH dataset to serve as the backbone to create similar questions. For each similarity level defined in Section 3.1, each backbone question is extended into a set of 10 (20 for math) questions, resulting in a final dataset with 1500 questions. Examples of questions with different similarities are given in Appendix E. The procedure of creating similar questions based on backbone questions is given in Appendix F.

**Memory Methods.** In this study, we explore both parametric-form and textual-form memory (Zhang et al., 2024c). In addition to the baseline of no memory mechanism, we conduct experiments to evaluate five memory methods: one parametric method (SFT) and four text-based memory methods.

- **No Memory** (Baseline): Questions are processed individually without memory mechanism.
- **SFT** (Supervised Fine-Tuning) as memory (Shao et al., 2023; Wang et al., 2023b; Yang et al., 2023a): The past question and generated answer pairs are used as data to perform supervised fine-tuning on the model.
- **In-Context** (Zhao et al., 2024; Huang et al., 2023a; Wang et al., 2023a): The past questions and generated answers are used as in-context examples to guide the reasoning.
- **Reflection**: LLMs self-summarize past experiences and summarize rules to guide the reasoning. We consider three variants: 1) **Reflect** (individually) (Zhong et al., 2024; Packer et al., 2023): The LLM individually reflects on each previous question and answer pair and summarizes experience from each one. 2) **Multi-Case Reflect** (Zhao et al., 2024; Tack et al., 2024): The LLM is asked to summarize experience from multiple previous question and answer pairs. 3) **Reflect-Update** (Hu et al., 2023; Shinn et al., 2023): The LLM maintains a reflection by updating it with each new question and answer pair.

**Metrics.** We evaluate 1) allocated *compute budget* and 2) *accuracy* on each question. Accuracy is evaluated towards the ground truth answer for math, commonsense and logical reasoning tasks, and pass rate against the test code for coding tasks. For each single question, these two metrics are averaged over four runs.

**Implementation Details.** Experiments on Best-of-N, DFS, and Self-Refine are conducted on the Llama-3.1-8B model, and experiments on Long CoT are conducted on the DeepSeek-R1-Distill-Qwen-7B model. For score estimation, we use `gpt-4o-mini` as the scorer model. For all LLMs, we use a temperature of 0.7 and top_p of 0.9 in generation. The data and code are available at `https://anonymous.4open.science/r/llm_efficiency_self_improve-7CD9/`. More implementation details are given in Appendix G.

## 5 RESULTS

We show the compute budget trends across different levels of question similarity, reasoning strategies, and memory methods in math reasoning in Fig. H.1. Then we discuss the reasoning speedup behaviour along different axes in the following subsections.

### 5.1 POSSIBILITY OF LLM REASONING SPEEDUP

***Finding 1: LLMs can generally achieve reasoning speedup through past experience.***

We first show that such reasoning speedup behavior is generally observed across various reasoning tasks and task similarities (Fig. 1), as well as memory methods and inference methods (Fig. 2). As shown in Fig. 2, the left panel demonstrates significant reductions in compute budget on the math dataset, with up to a 56% reduction when using the combination of *In-Context* memory and *DFS* reasoning in the similarity level S3, and there is at least a 10% reduction in 47.5% settings. Additionally, as shown in Fig. 1, reasoning speedup consistently occurs in all four types of tasks in our experiments, with complex reasoning tasks such as math reasoning having the highest efficiency gain, demonstrating that reasoning speedup is a general behavior for LLMs when equipped with memory and adaptive compute allocation.

***Finding 2. Reasoning efficiency gains increase with question similarity.***

We also find that reasoning speedup is more pronounced when questions are more similar. This effect can be highlighted from Fig. 1, where results from four reasoning tasks are grouped and averaged by their similarity levels. As shown, the reductions in compute budget are most significant in more similar question groups (S1 and S2). As more details are shown in Fig. 2 (left), such pattern is consistent across most memory methods, especially *SFT*, *In-Context*, and *Reflect-Update*. This pattern is largely expected, since it aligns with patterns observed in human cognition, where efficiency improves more for more similar tasks.

***Finding 3. Response speed and accuracy are strongly correlated; faster responses tend to be more accurate.***

Fig. 2 shows a clear correlation between reasoning efficiency (compute budget) and accuracy. Specifically, regions with deeper blue in the left panel (more reductions in compute cost) often correspond to regions with deeper red in the right panel (more accuracy improvements). Upon examination, the relative compute budget and accuracy have a Pearson correlation of -0.41 with p=0.0002, suggesting a moderate and statistically significant negative correlation. These observations suggest that enhancing reasoning speed does not sacrifice, but instead improves answer correctness. This is because test-time scaling methods suffer from the gap between the estimated answer quality score and actual correctness, but with memory with verified correct answers, this issue can be alleviated, leading to improved accuracy.

### 5.2 THE EFFECTS OF MEMORY METHODS AND INFERENCE METHODS

***Finding 4. Episodic memory methods generally outperform semantic memory methods in LLM reasoning speedup.***

In Fig. 3, we show the relative compute budget values of each memory method, averaged across different similarity levels and test-time scaling methods in the math dataset. Generally, we find that episodic memory methods (*SFT*: 10.8%, *In-Context* 27.4%) reduce compute budgets more effectively than semantic (reflection-based) methods (3.6%, 5.5%, 8.8%). This aligns with previous studies showing that comprehensive recall of past experience is important for benefiting LLMs in problem solving (Renze & Guven, 2024). Similarly, psychological research indicates that human proficiency initially relies on episodic memory, which allows for instance-based retrieval (Logan, 1988).

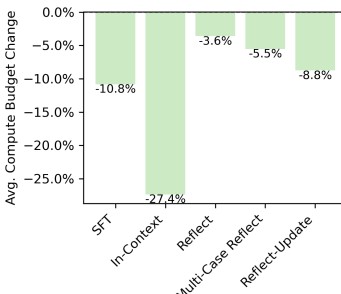

Figure 3: Compute budget changes grouped and averaged by memory methods.

***Finding 5. Text-based memory methods are more effective for shorter-term reasoning speedup, while parametric memory is more suited for longer-term reasoning speedup.***

In Fig. 4, we show the relative compute budget averaged across different similarity levels and test-time scaling methods. The results reveal that memory methods relying on in-context textual memory, including *In-Context*, *Reflect*, and *Multi-Case Reflect*, achieve efficiency gains more efficiently in the short term. This is because the nature of ICL outperforms SFT in few-shot settings Luo et al. (2023); Zhang et al. (2024a), but the efficiency gain is further limited by the context length issue of LLM architectures. Note that while modern LLMs can handle contexts up to 1M tokens, we cap the length at 3 memories to highlight this behavior. In contrast,

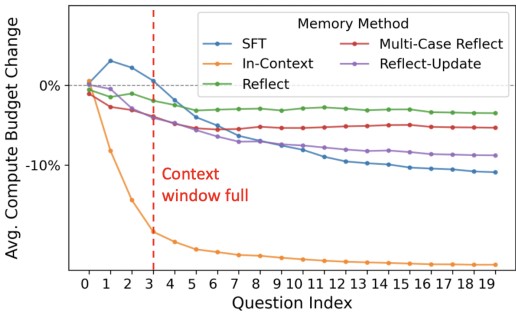

Figure 4: Compute budget change in each question index, grouped and averaged by memory methods.

*SFT*, which saves memory into the model parameters, shows more consistent efficiency gains across the entire sequence, which has the potential for better suiting longer-term reasoning speedup. As shown in Fig. 4, after approximately 10 questions, SFT surpasses the reflection-based methods in efficiency gains.

***Finding 6. The effect of different test-time scaling methods is correlated with memory methods.***

Based on our results, there is no single best test-time scaling method for enabling LLM reasoning speedup. As shown in Fig. 2, the effectiveness of scaling methods is closely related to the type of memory method used. When using episodic memory methods such as *SFT* and *In-Context*, *DFS* appears to be the most effective in reducing compute budgets. In contrast, when using reflection-based memory methods, *Self-Refine* and *Long CoT* yield better efficiency gains. Given the strong influence of the scaling method on the overall LLM performance, which is also evidenced by the fact that many state-of-the-art models such as OpenAI o1 (Jaech et al., 2024) and DeepSeek-R1 (Guo et al., 2025) adopt *Long CoT*, it is important to recognize that the choice of scaling methods should not be the first consideration. However, for *Long CoT*, we do observe that different memory methods show similar efficiency improvements, with *In-Context* performing slightly better.

### 5.3 FAILURE CASE ANALYSIS

While memory mechanisms generally reduce compute costs for similar questions, in experiments we found that under certain conditions memory still can increase compute costs and degrade accuracy, such as in Fig. 2, warm cells in the left panel and cool cells in the right panel. Across all the results, we observed that such an efficiency drop primarily occurs in low similarity questions (such as S4 in math reasoning). This relates to the assumption in Theorem 2, that memory from similar questions does not hurt performance on query tasks. When the questions and answers in memory differ substantially from the current query, the model can overfit to irrelevant examples from memory (Zhang et al., 2024b), and repeated reliance on a narrow set of memories can trigger catastrophic forgetting, reducing the model's ability to generalize (Luo et al., 2023), and in these cases the assumption in Theorem 2 does not hold. This motivates future research on effectively deciding the memory to store or retrieve for the reasoning speedup behaviour.

## 6 CONCLUSION

In this study, we raised and formally defined the question of whether large language models (LLMs) can achieve reasoning speedup through repeated exposure. To address this, we proposed SPEEDU-PLLM, a unified framework for implementing and benchmarking reasoning speedup behaviors in LLMs. Through extensive experiments, we observed that reasoning speedup generally emerges across different reasoning tasks, memory mechanisms and test-time scaling methods, particularly when the questions exhibit higher similarity. Additionally, we provided several insights into the factors that influence such behaviors.

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

# A  ADDITIONAL RELATED WORK

## A.1  LLM EFFICIENT REASONING

Although test-time scaling significantly boosts LLMs' reasoning ability, it also results in substantial computational overhead and increased reasoning time (Sui et al., 2025). Therefore, various types of LLM efficient reasoning methods have been developed. Existing methods can be categorized into RL-optimization with length reward (Luo et al., 2025; Aggarwal & Welleck, 2025; Team et al., 2025), SFT with shorter CoT (Yu et al.; Kang et al., 2025; Liu et al., 2025), latent representation compression (Hao et al., 2024; Cheng & Van Durme, 2024; Xu et al., 2025b), dynamic reasoning algorithms (Sun et al., 2024; Wang et al., 2025; Ding et al., 2025), prompt-guided efficient reasoning (Han et al., 2024; Lee et al., 2025; Xu et al., 2025a), training data efficiency methods (Ye et al., 2025; Muennighoff et al., 2025), model compression and distillation (Sun et al., 2025; Zhang et al., 2025a). To our best knowledge, no existing research has focused on exploring the efficiency brought by memory or exposure to similar questions.

# B  TASK SIMILARITY DEFINITIONS

Below are the definitions of task similarities in commonsense reasoning tasks (Table 2), code generation tasks (Table 3), and logical reasoning tasks (Table 4).

Table 2: Definition of question similarity levels in commonsense reasoning.

| Level | Description | Example |
|-------|-------------|---------|
| S1 | Exactly the same questions. | "Why do people wear coats in winter?" vs "Why do people wear coats in winter?" |
| S2 | Same questions, different wording | "Why do people wear coats in winter?" vs "What is the reason people put on coats when it's cold outside?" |
| S3 | Same underlying knowledge, different specific questions. | "Why do people wear coats in winter?" vs "Why do people turn on heaters when the weather gets cold?" |

Table 3: Definition of question similarity levels in code generation.

| Level | Description | Example |
|-------|-------------|---------|
| S1 | Exactly the same questions. | "Write a Python function to reverse a string." vs "Write a Python function to reverse a string." |
| S2 | Same questions, different wording | "Write a Python function to reverse a string." vs "How can I implement a function in Python that takes a string and outputs its reverse?" |
| S3 | Same answer structure, different specific questions. | "Write a Python function to reverse a string." vs "Write a Python function to check if a word is a palindrome." |

# C  PROOF OF THEOREMS AND COROLLARY

## C.1  PROOF OF THEOREM 1

*Proof.* From the assumptions we have:

$$\mathbb{P}\left(\max_{1 \leq j \leq k} \texttt{score}(r_j^{(t+1)}) \geq \tau\right) \geq \mathbb{P}\left(\max_{1 \leq j \leq k} \texttt{score}(r_j^{(t)}) \geq \tau\right).$$

Table 4: Definition of question similarity levels in logical reasoning.

| Level | Description | Example |
|-------|-------------|---------|
| S1 | Exactly the same questions. | "If all cats are animals and Max is a cat, is Max an animal?" vs "If all cats are animals and Max is a cat, is Max an animal?" |
| S2 | Same questions, different wording | "If all cats are animals and Max is a cat, is Max an animal?" vs "Given that every cat belongs to animals, and Max is a cat, does Max count as an animal?" |
| S3 | Same structures, different specific questions. | "If all cats are animals and Max is a cat, is Max an animal?" vs "If all roses are flowers and Daisy is a rose, is Daisy a flower?" |

So the cumulative distribution of $|\mathcal{R}_{s'}^{(t)}|$ satisfies:

$$\mathbb{P}\big(|\mathcal{R}_{s'}^{(t)}| \leq k\big) = \mathbb{P}\left(\max_{1 \leq j \leq k} \mathrm{score}(r_j^{(t)}) \geq \tau\right).$$

Since $\mathrm{cost}(\mathcal{R})$ is non-decreasing with $|\mathcal{R}|$, the expected cost satisfies

$$\mathbb{E}\big[\mathrm{cost}(|\mathcal{R}_{s'}^{(t+1)}|)\big] \leq \mathbb{E}\big[\mathrm{cost}(|\mathcal{R}_{s'}^{(t)}|)\big].$$

$\square$

## C.2 Proof of Theorem 2

*Proof.* Let $M_k^{(t)} = \max_{1 \leq i \leq k} \mathrm{score}(r_{i;s}^{(t)})$. We show $M_k^{(t+1)} M_k^{(t)}$.

**Under (C1) Independence:** Let $p_{j,t} = \mathbb{P}(\mathrm{score}(r_{j;s}^{(t)}) < \tau)$. The dominance assumption implies $p_{j,t+1} \leq p_{j,t}$. The probability of failure (all $k$ below $\tau$) is $\mathbb{P}(M_k^{(t)} < \tau) = \prod_{j=1}^{k} p_{j,t}$. Since every term in the product decreases or stays same, the total probability of failure decreases. Thus, the probability of success increases.

**Under (C2) Monotonic DAG:** We induct on the generation order. Roots improve by the premise. For any node $j$, if parents $\mathcal{P}(j)$ improve stochastically, the Monotonicity condition implies child $j$ improves stochastically. Thus, the joint distribution of the first $k$ scores at $t+1$ stochastically dominates that at $t$. Since $\max(\cdot)$ is a monotonic function, the dominance is preserved for the maximum value. $\square$

## C.3 Proof of Corollary

*Proof.* Immediately with Theorem 1 and 2. $\square$

# D  Test-time Scaling Methods

In this appendix, we elaborate how several widely-used test-time scaling methods can be expressed within the unified framework introduced in Section X. Let $f_s(q^{(t)})$ denote the set of candidate answers $\mathcal{R}_s^{(t)}$ generated by applying test-time scaling method $s \in \mathcal{S}$ to question $q^{(t)}$. Each method defines a minimal unit of computation, a corresponding cost function $\mathrm{cost}(\cdot)$, and a score function $\mathrm{score}(\cdot)$ to evaluate answer quality.

**Best-of-N.** This method generates $N$ complete candidate answers. Each $r_{k;s}^{(t)} \in \mathcal{R}s^{(t)}$ is an independently sampled answer.

- *Unit of computation:* full answer.

- *Cost:* number of generated answers, i.e., $|\mathcal{R}s^{(t)}|$.

- *Score:* LLM Judge or Process Reward Model assigns $\text{score}(r)$ for each candidate $r$.

- *Final answer:* $\text{argmax}\, r \in \mathcal{R}s^{(t)}\, \text{score}(r)$.

This method directly aligns with our unified formulation, with $f_s$ sequentially or in batches generating $r1, r2, \ldots, r_N$ until an answer satisfying the threshold is found.

**Tree-of-Thoughts.** This method performs structured reasoning by expanding nodes in a search tree.

- *Unit of computation:* tree node.

- *Cost:* number of generated and scored nodes.

- *Score:* each node is scored via LLM Judge or reward model.

- *Final answer:* the highest-scoring reasoning path.

Our formulation covers this by letting $f_s$ denote the expansion of search tree nodes, with $\mathcal{R}_s^{(t)}$ representing partial reasoning paths. Both DFS and BFS are special cases, differing in node expansion order.

**Self-Refine.** This method iteratively generates improved answers based on previous attempts.

- *Unit of computation:* full refined answer.

- *Cost:* number of refinement steps (full answers).

- *Score:* internally assessed (e.g., by the model itself) or by an external judge.

- *Final answer:* the most refined answer exceeding the threshold.

Self-Refine is inherently adaptive: $f_s$ produces a sequence of revised answers until $\text{score}(r) \geq \tau$. No explicit compute optimization is required.

**Long CoT (Chain-of-Thought).** This method enables free-form, unbounded reasoning over long text segments.

- *Unit of computation:* token.

- *Cost:* number of generated tokens.

- *Score:* assessed implicitly by the model (e.g., via stopping probabilities).

- *Final answer:* the whole generated sequence.

This method integrates self-refinement and tree-search behavior into token-level generation. $\mathcal{R}_s^{(t)}$ here can be seen as the full text trace, and $\text{score}(\cdot)$ may correspond to internal confidence or the model's own stop criterion (e.g., omitting "wait" tokens).

# E  DATA EXAMPLES

**Question Backbone**:

```
Let $\mathbf{a} = \begin{pmatrix} 1 \\ 1 \\ 0 \end{pmatrix}$ and $
   \mathbf{b} = \begin{pmatrix} 2 \\ 0 \\ -1 \end{pmatrix}.$
   Find the vector $\mathbf{v}$ that satisfies $\mathbf{v} \times
   \mathbf{a} = \mathbf{b} \times \mathbf{a}$ and $\mathbf{v} \
   times \mathbf{b} = \mathbf{a} \times \mathbf{b}.$
```

**S1**: (Exactly the same with the question backbone.)

**S2** (Same numbers, different wording):

```
Given the vectors $\\mathbf{a} = \\begin{pmatrix} 1 \\\\ 1 \\\\ 0
   \\end{pmatrix}$ and $\\mathbf{b} = \\begin{pmatrix} 2 \\\\ 0
   \\\\ -1 \\end{pmatrix}$, determine the vector $\\mathbf{v}$
   that meets the conditions $\\mathbf{v} \\times \\mathbf{a} =
   \\mathbf{b} \\times \\mathbf{a}$ and $\\mathbf{v} \\times \\
   mathbf{b} = \\mathbf{a} \\times \\mathbf{b}$.
```

**S3** (Same structure, different numbers):

```
Let $\\mathbf{a} = \\begin{pmatrix} 0 \\\\ -3 \\\\ -2 \\end{
   pmatrix}$ and $\\mathbf{b} = \\begin{pmatrix} -1 \\\\ -3
   \\\\ 0 \\end{pmatrix}.$  Find the vector $\\mathbf{v}$
   that satisfies $\\mathbf{v} \\times \\mathbf{a} = \\mathbf
   {b} \\times \\mathbf{a}$ and $\\mathbf{v} \\times \\mathbf
   {b} = \\mathbf{a} \\times \\mathbf{b}.
```

**S4** (Same underlying knowledge, different structure and numbers):

```
Find the scalar $k$ such that the points $(1, k)$, $(k, 2)$,
   and $(3, 4)$ are collinear.
```

## F  DATASET CONSTRUCTION

Starting from a backbone question, we first prompt an LLM to generate similar questions, and then ensure correctness using either programmatic verification or strong LLMs. Specifically, we use GPT-o3 to generate candidate questions with prompts that specify the similarity definition (e.g., "reword the question," "generate questions based on the same underlying knowledge"). We then verify the correctness of the answers before adding them to the dataset. For math tasks, we employ program-based verification: given a candidate question, we calculate the correct answer using code execution. For commonsense and logical reasoning, we rely on a strong LLM (GPT-5 reasoning) to generate answers, which are then verified by humans. For coding tasks, we use GPT-5 reasoning to generate test code corresponding to the candidate questions, followed by human verification of the test cases.

## G  IMPLEMENTATION DETAILS

We develop a unified test-time reasoning framework in Python, using PyTorch and Hugging Face's Transformers. Our experiments are conducted on machines with NVIDIA A100, H100, H200, L40S and L4 GPUs, with a total of 4000+ GPU hours.

**Test-Time Scaling Methods.**  Below, we detail the core mechanisms, configurable parameters, and termination criteria:

- **Best-of-N**: Generates $N$ candidates (default = 5). Each is scored using a value model or PRM, and a candidate is selected based on value and correctness. *Termination:* stops early if a candidate meets the score threshold; otherwise, completes all $N$ evaluations.

- **Self-Refine**: Begins with a generated candidate and iteratively improves it using feedback and refinement prompts. We set the max refinement steps as 15. *Termination:* stops early if the feedback indicates "No error"; otherwise continues until the max iteration count is reached.

- **Long CoT**: A single long-form reasoning trace is generated, typically prefixed with a `<think>` tag. The model self-determines when to stop generating, often marked by the token `</think>`. *Termination:* when the model stops generating or produces an explicit termination tag. We set a max token number of each answer as 3500 tokens.

- **Depth-First Search (DFS)**: A search tree is constructed over reasoning steps, where nodes represent partial reasoning segments. Child nodes are generated up to `max_depth` (default = 15), and evaluation is guided by value thresholds (`value_thresh`) and pruning ratio

(0.4). *Termination:* when either a termination node (the content has "End of Answer" with perfect value (1.0) is found, or the search exceeds `max_node` (default = 50) expansions, or all candidates are exhausted.

**Memory Methods.** We implement the following memory mechanisms.

- **No Memory**: The baseline configuration with no carry-over between rounds.
- **Supervised Fine-Tuning (SFT)**: The model is updated after each question using a single-step gradient descent. For reasoning models (e.g., `DeepSeek-R1`), we generate a compressed representation of the reasoning trace using a summarization prompt. The fine-tuning is performed using the following hyperparameters: `learning_rate` = 5e-4, Single-step update with LoRA enabled. The SFT method is only applied if a correct and high-quality answer is found.
- **In-Context**: A memory buffer maintains the last $n$ successful examples. For each new prompt, we prepend these examples as demonstrations. We set the maximum number of in-context examples to 3.
- **Reflection**: After a correct response, a language model is prompted to reflect on the reasoning process. The resulting reflection is stored and used in future prompts under the "Consider:" section. Only one reflection is stored per example.
- **Multi-Case Reflection**: Instead of prompting reflection on a single instance, this method generates a joint reflection across multiple past successful cases. All stored examples are included in a joint input to generate an abstracted reflection. We limit the context by a maximum of 3 examples.
- **Reflect-Update**: This method iteratively refines a single running reflection. After each correct answer, the previous reflection and the new case are used to generate an updated reflection. The updated reflection replaces the old one, maintaining a compact and evolving summary of reasoning strategies.

**Memory Update Policy.** Memory is updated only if a qualifying answer is found during the current question's rounds. In the four tasks in our experiments, the correctness is checked with the ground truth. In scenarios when the ground truth is not available, this can also be implemented with an LLM-Judge.

**Evaluation Configuration.** We run the evaluation over 10 question sets per dataset, with 4 repetitions for each question, to report the mean value.

**Scoring.** We use gpt-4o-mini as the scorer model. The score threshold is set as $\tau = 1.0$, which in our experiments is implemented as the scorer model (gpt-4o-mini) decided this answer (or one reasoning step) is "surely correct" among three options of "surely correct", "maybe correct", or "surely incorrect".

# H ADDITIONAL RESULTS

## H.1 FULL RESULTS ON MATH DATASET

See Fig. 5.

## H.2 EXAMPLES OF DIFFERENT REFLECTION METHODS

Here we show the reflection generated from the same question backbone.

**Reflect (individually):** (**This is an unrelated reflection derived from a less-similar question in the same set with Similarity S4.**)

```
 - Identify the principal amount ($1,000), the interest rate (5%),
and the time period (3 years).
 - Use the formula for simple interest:  Interest = Principal ×
```

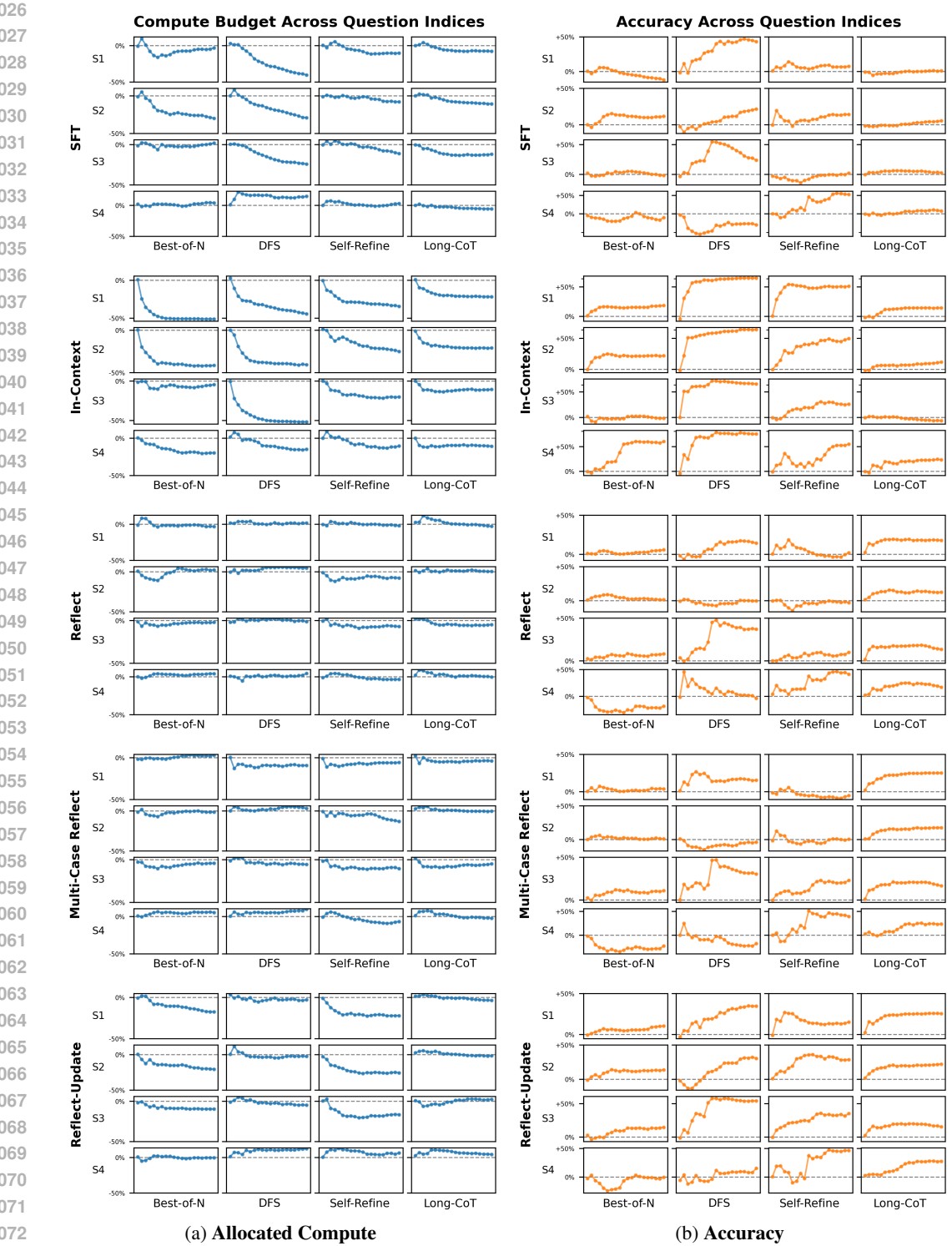

(a) **Allocated Compute**          (b) **Accuracy**

Figure 5: Changes in (a) compute budget and (b) accuracy relative to the baseline (no memory), across different memory methods, test-time scaling methods, and question similarity levels in the math dataset. In each subplot, the x-axis denotes the question index within a sequence of questions, while the y-axis shows the percentage change in compute budget or accuracy compared to the baseline. Each curve presents values averaged over multiple question sets at each question index. Gray dashed lines represent baseline (no memory) performance. Results show that the reasoning speedup behaviour is general across different settings. Results on other tasks are given in Appendix H.3.

```
Rate × Time.
- Substitute the known values into the formula:  Interest = $1,000
× 0.05 × 3.
- Calculate the interest:  Interest = $150.
End of answer.
```

**Multi-Case Reflect**:

```
 - Define the initial positions of the train and car as (0, 0).
- Let the speed of the train be $v_t$ and the speed of the car be $v_c$.
- After time $t$, the position of the train will be $(0, v_t \cdot t)$ and the
position of the car will be $(v_c \cdot t, 0)$.
- Use the distance formula $d = \sqrt{(x_2 - x_1)^2 + (y_2 - y_1)^2}$ to express the
distance $D$ between the train and car as:
```

$$D = \sqrt{(v_c \cdot t - 0)^2 + (0 - v_t \cdot t)^2} = \sqrt{(v_c \cdot t)^2 + (v_t \cdot t)^2}$$

```
- Simplify the equation to:
```

$$D = t \cdot \sqrt{v_c^2 + v_t^2}$$

```
- Substitute the known values of $t$, $v_c$, and $v_t$ to find the distance
$D$ after the specified time.
End of answer.
```

**Reflect-Update:** x

```
 - Reaffirmed the use of the centripetal acceleration formula
$a_c = \frac{v^2}{r}$ as a fundamental principle in circular motion.
- Clarified the substitution of values, ensuring each step is
transparent and logically follows from the previous one, which
aids in comprehension.
- Highlighted the importance of checking calculations at each
stage, particularly in deriving $a_c$ and converting to terms of $g$.
- Emphasized the need for consistent units throughout the
calculation to avoid errors, reinforcing unit conversion
practices.
- Improved the rounding approach to maintain precision in the
final result, ensuring it aligns with significant figures based
on input values.
- Incorporated a systematic verification step to cross-check
the final answer against physical principles, ensuring it is
reasonable and accurate.
- Continued to prioritize expressing results in terms of $g$,
fostering a deeper understanding of the practical implications
of centripetal acceleration in real-world scenarios.
End of answer.
```

H.3   RESULTS ON COMMONSENSE REASONING, CODING AND LOGICAL REASONING TASKS

We first evaluated all five memory methods and four test-time scaling methods on math reasoning tasks. Then, to evaluate the generalizability of the conclusions on math tasks and meanwhile control the total cost, we conducted additional experiments on three other tasks, primarily focusing on representative settings we identified from math tasks, which are DFS and Long CoT for reasoning, and SFT and In-Context for memory.

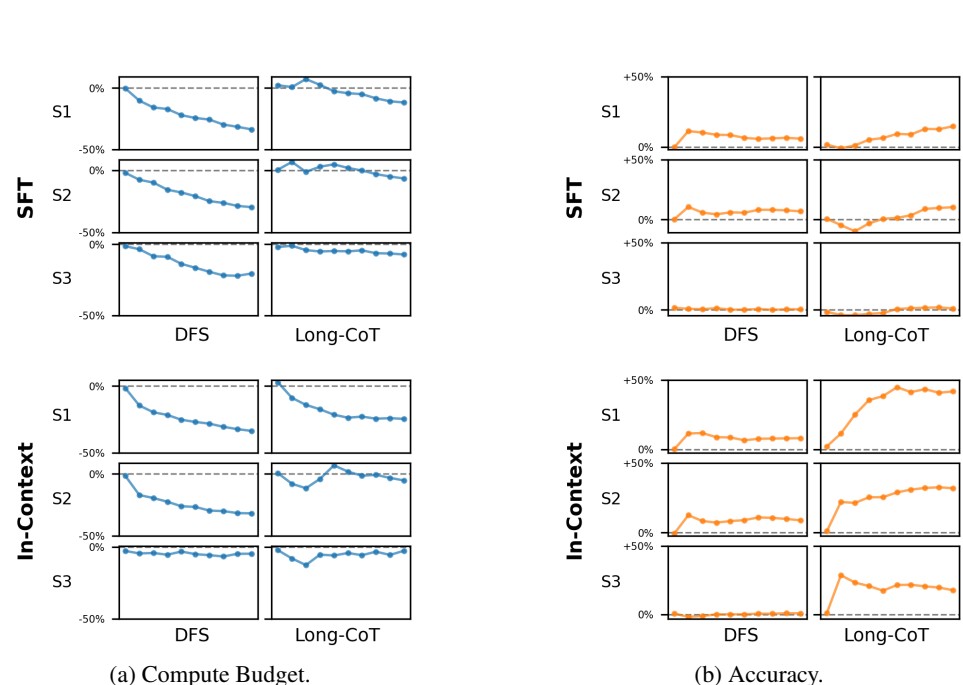

(a) Compute Budget.

(b) Accuracy.

Figure 6: Results on COMMONSENSEQA.

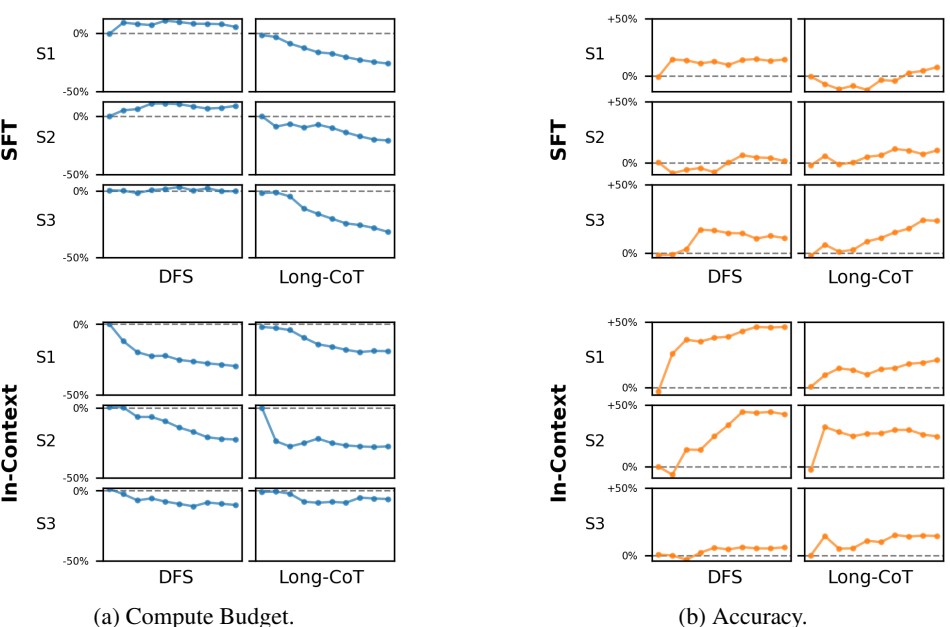

(a) Compute Budget.

(b) Accuracy.

Figure 7: Results on PRONTOQA (Logical Reasoning).

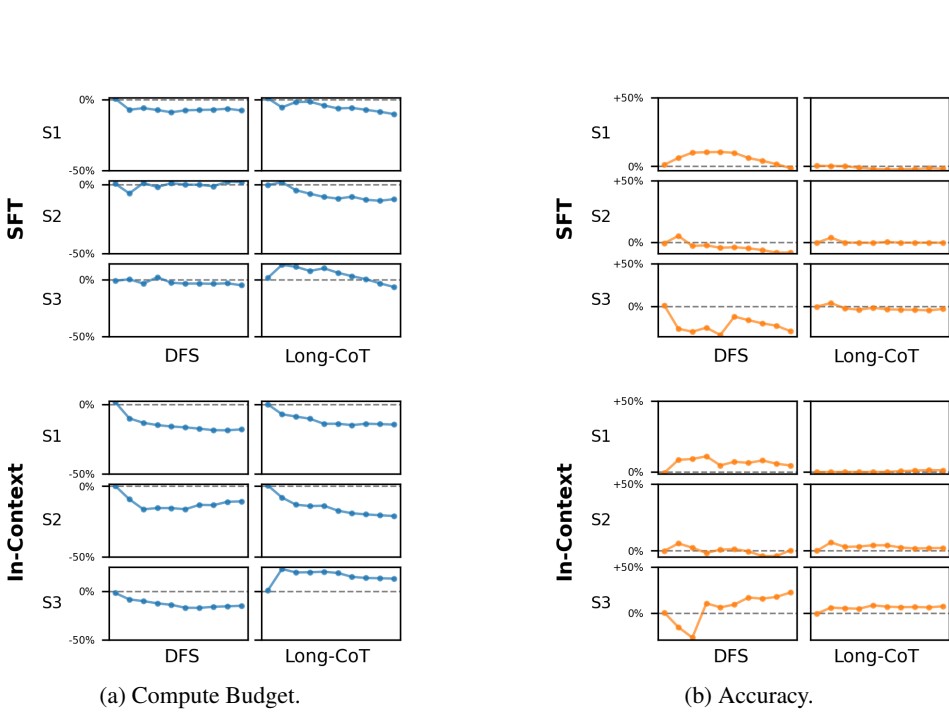

(a) Compute Budget.

(b) Accuracy.

Figure 8: Results on HUMANEVAL (Coding).

# I    STUDY ON REAL-WORLD DATASET

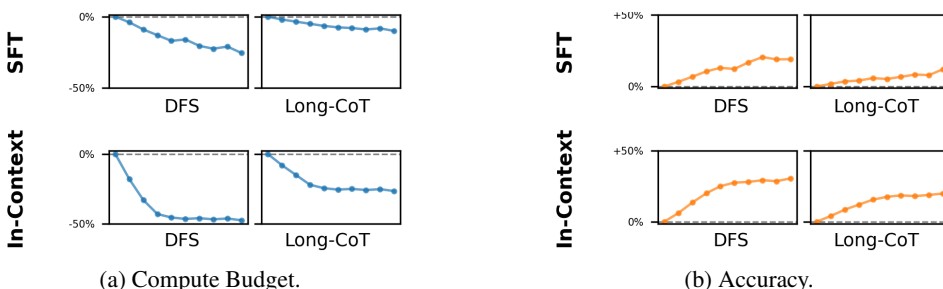

(a) Compute Budget.                    (b) Accuracy.

Figure 9: Results on Stack Overflow (Real-World Study).

To study the proposed framework on a real-world dataset, we create an additional test set based on the Stack Overflow database. The Stack Overflow dataset has a label for duplicate questions. We extract a test set of 20 clusters, each cluster with 10 questions marked as duplicated by humans. A strong LLM, GPT-5, is used to evaluate the answer correctness provided with the reference answer (with the highest score on Stack Overflow). Results show that our proposed framework can consistently improve reasoning efficiency in real-world QA streams.

