# OpenReview forum: "Can Past Experience Help LLMs Reason Faster?"
_ICLR.cc/2026/Conference — Submitted to ICLR 2026_

### Official Review · Reviewer_iSMS · 2025-10-28

**Soundness:** 2
**Presentation:** 1
**Contribution:** 2
**Rating:** 4
**Confidence:** 4

**Summary:**

This paper aims to study how to speedup LLM reasoning through both test-time compute allocation and memory mechanism. The authors empirically investigated diffrent test-time scaling methods and memory mechanisms, showing that past experience (i.e., memory) could speed up reasoning and it is dependent on the test-time scaling method and task similarity.

**Strengths:**

- Considering the synergy between test-time scaling and memory mechanism is interesting.
- The studied test-time scaling methods and memory mechanisms are comprehensive.

**Weaknesses:**

- The presentation of this paper could be further improved to highlight its significance and strengthen its clarity. For example, it is hard to understand whether the authors try to propose a new algorithm or just analyze/study the speedup problem. In addtion, the combination of adaptive test-time allocation and memory mechanism is not well-justified. From the title itself, it seems the focus of this paper should be investigating the memory mechanism. It might be clearer if the authors could first show speedup LLM reasoning through test-time scaling is dependent on memory mechanism.
- The theorems are straightforward and not very meaningful. Especially, the assumptions (e.g., the probability that the best response exceeds the quality threshold is non-decreasing with t) in both theorem 1 and theorem 2 are unrealistic. If consider Best-of-N case with independent sampling, the probability remains the same, which would give a trivial result. Besides, the assumption in theorem 2 assumes that including more memory (e.g., in-context) is always good, but in reality, this is often violated, as the model will eventually suffer from longer context.
- The task similarity is not technically formulated well, and it is unclear how task similarity could be leveraged in the proposed framework, as the experimental discussions are only for different task similarity levels. I would expect that, given some task similarity information, the framework could optimize either the compute allocation or improve the memory mechanism.

**Questions:**

See weaknesses above.

---

> ### Author Response · Authors · 2025-11-29
>
> Dear Reviewer iSMS,
>
> We appreciate your taking time to review our article and your insightful comments. We want to address your concerns here.
>
>
> **(W1A: whether try to propose a new algorithm or study the speedup problem)**
>
> **Response:** We thank the reviewer for raising this point. Our primary goal is to **study the reasoning speedup phenomenon**. To study it, there are two **barriers** that prevent current LLM systems from exhibiting speedup, and therefore **introduce a minimal framework** that integrates (i) adaptive test-time allocation and (ii) memory mechanisms to make such speedup observable and comparable across paradigms.
>
> We have **revised the Methodology** section to make this focus and separation clearer.
>
>
> **(W1B Justification for combining adaptive allocation and memory)**
>
> **Response:** We **revised the draft to clarify that the two components** have distinct and complementary roles: **memory** increases the likelihood that high-quality candidates appear early; **adaptive allocation** converts this improvement into reduced compute. **Neither component alone yields speedup.**
>
> **W2. Theorem not meaningful**
>
> **General response:** We believe there was a misunderstanding of the purpose of the theoretical section.
> The theorems are **not intended to claim that speedup always happens**.
> Instead, they provide a decomposition of the **necessary conditions under which speedup is possible**:
>
> Theorem 2: Under mild assumptions, relevant memory increases the probability of early good candidates.
>
> Theorem 1: When this improvement occurs, adaptive allocation yields non-increasing compute.
>
> Together these two theorems identify when speedup can occur, instead of stating speedup will always occur.
>
>
> **(W2A: Trivial Results under Best-of-N sampling)**
>
> **Response:** We respectfully point out that this is a misunderstanding of our theorem. The assumption in Theorem 1 refers to the setting where memory is allowed to influence answer quality. We are comparing the **answers with and without memory, instead of different answer samples generated under the same condition**. So even for independent sampling, it is still a non-trivial result.
>
> **(W2B: Assumption not always hold.)**
>
> **Response:**  Theorem 2 is **intentionally stated with explicit conditions**.
>
> These conditions capture the case where memory is relevant and beneficial, and we are careful to state that they are not always satisfied (see our discussion at the end of Section 3.2). The theorem is a conditional result, **not an unconditional claim that “memory is always good”**. Actually, in our results part, we also connected the empirical result of efficiency drop under low similarity with this assumption.
>
> We hope the above clarifications can address your concerns.

---

### Official Review · Reviewer_n6qX · 2025-10-30

**Soundness:** 4
**Presentation:** 3
**Contribution:** 3
**Rating:** 8
**Confidence:** 3

**Summary:**

This paper investigates the effect of repeated exposure to similar questions on reasoning speed in LLMs. They focus on adaptive compute-budget allocation methods, where the cost of reasoning is reduced by early stopping. Additionally, they look at several methods of incorporating previous questions in memory, such as SFT, in-context learning, and reflection.

They propose a theoretical framework to unify several different types of adaptive test-time scaling methods and memory mechanisms. They show that if the memory mechanism does not decrease answer quality, then SpeedUpLLM achieves non-increasing budget while maintaining answer quality. Finally, they benchmark the method across tasks, question similarity, memory mechanisms, and test-time scaling mechanisms.

**Strengths:**

The paper explicitly outlines its contribution (p2). It is well-situated in prior work on test-time scaling and memory methods. When the theoretical framework is outlined, the authors specifically describe how various common test-time scaling methods can be made adaptive, which makes the description concrete.

The experiments are extensive and show clear trends across the different axes that are evaluated. Further, the results are broken into several ‘findings’ sections, where the authors discuss the implications and significance of each observation in detail. I also appreciated how comparisons were made to results in human cognition.

**Weaknesses:**

I don’t see any significant weaknesses with this work.


My only suggestion is that Figures 1 and 3 are difficult to parse. Particularly with Figure 1, it is not easy to compare across the different variables by looking at it. Additional figures like Figure 5 for scaling types or question similarity would be useful.

**Questions:**

The theoretical analysis requires that memory ‘does not degrade model performance’. Do you think that methods like ‘Reflect’ didn’t scale as well in the long horizon for this reason, or is it because the reflections don’t have as much capacity to hold information?

I would expect the LMs to be completely accurate on the S1 type question, since the answer to the identical question is included in memory (I guess specifically for the in-context mechanism). Why is this not the case?

Typos:

Theorem 2 – …the probability [that] a satisfying answer appears…

---

> ### Author Response · Authors · 2025-11-29
>
> Dear Reviewer n6qX,
> We appreciate your taking time to review our article and your insightful comments. We want to address your concerns here.
>
> **W1 (Figure Clarity)**
>
> **Response:** Thank you for the suggestion. We have updated Figures 1 and 3 by adding clearer, concise takeaways in the captions.
>
> **Q1.**
>
> **Response:** The degradation of methods like Reflect at longer horizons is primarily due to the abstractness and limited specificity of the reflections. Reflection stores summarized insights rather than concrete problem&solution information, which limits its ability to support precise reuse for later questions.
>
> **Q2.**
>
> **Response:** This is because although S1 contains identical questions, not all such questions are answerable by the base LLM. For difficult items where the model consistently produces incorrect answers, memory cannot “repair” the model’s underlying capability limitations. Since S1 accuracy is averaged across a set of identical-question clusters, these inherently hard cases lower the aggregate accuracy and prevent it from reaching 100%.
>
> **Typo:**
> **Response:** We have fixed the typo. Thanks for pointing it out.
>
> We hope the above clarifications can address your concerns.

---

### Official Review · Reviewer_ueHS · 2025-11-03

**Soundness:** 3
**Presentation:** 2
**Contribution:** 3
**Rating:** 4
**Confidence:** 3

**Summary:**

The paper studies whether LLMs can “reason faster with experience.” It proposes SPEEDUPLLM, combining (i) adaptive compute allocation (early stopping under a score threshold τ) and (ii) a memory mechanism (SFT or text memories such as in-context and reflections). Theoretical results (Thm. 1–2) claim that as answer quality improves with experience, expected compute decreases; and that expanding relevant memory does not hurt the probability of producing a satisfactory answer within the first k candidates. Experiments across four task types report up to 56% compute reduction with comparable or higher accuracy when questions are similar.

**Strengths:**

1. Timely problem and clean framing of “reasoning speedup” by similarity levels (S1–S4) and compute budget definitions per decoding paradigm.

2. Unifies several test-time scaling methods under a simple early-stop view; clear to implement and evaluate.


3. Broad empirical sweep (tasks, scaling methods, memory variants) with some actionable insights (episodic memories > reflections; stronger gains at higher similarity; negative correlation between compute and error).

4. Public code link and sensible implementation details (models, scorer, temps).

**Weaknesses:**

Incremental significance – The main insight—that prior exposure and adaptive stopping can reduce compute—is intuitive and somewhat expected. The theoretical contribution would be more valuable if the assumptions were clearly formalized and the empirical validation extended to natural sequential data.

Theorem 1 clarity – The proof is difficult to follow. The assumption that cost(R) increases with |R| is never stated, nor is it clear whether |R⁽ᵗ⁾| is constant across t. These conditions are critical for the result and should be explicitly stated and justified.

Theorem 2 assumptions – The argument from expectation to higher probability is not rigorous. The theorem should be restated directly under a stochastic-dominance premise, with independence and DAG-monotonicity listed as explicit assumptions. The current proof is not convincing.

Modeling opacity – The theoretical framing (Eqs. 3–5) is overly compact and hard to interpret. It would help to provide small illustrative examples showing what cost, score, and adaptive mean in practice—e.g., that adaptive allocation simply finds a prefix of R⁽ᵗ⁾ whose score exceeds τ. Without such grounding, the modeling contribution feels abstract and under-explained.

τ selection and calibration – The method for choosing τ is unclear. Is it tuned per task, globally, or adaptively? Results should show sensitivity of both speedup and accuracy to τ and to the judge model used for scoring.

Memory degradation – The paper observes performance drops at low similarity but does not quantify when memory becomes harmful. Please report a retrieval-similarity threshold (e.g., embedding cosine or PRF) where memory is suppressed, and show an ablation.

Limited generalization – The evaluation is largely synthetic (similarity levels S1–S4). It remains unclear whether the findings hold on realistic, temporally ordered workloads such as near-duplicate QA streams under distribution drift.

**Questions:**

1. Theorem 1 details: i don't quite fully follow the proof. The assumption that cost(R) is increasing when |R| increases is never mentioned before. Also are you assuming |R^{t}| is a constant across t? Please state everything clearly.

2. Theorem 2 assumptions: Can you restate Thm. 2 with the stochastic-dominance premise directly (rather than mean-score), and list independence/DAG-monotonicity as explicit assumptions in the theorem body? I am not following the proof either.

3. Overall I think there is a value for modeling contribution in this paper. But it is too short for me to digest and appreciate. Can you provide examples/illustrations for exactly what cost, score, adaptive are etc. In particular, i feel the adaptive compute budget allocation is simply finding a prefix of R^{(t)} such that the score is large enough. The current formulations (3) - (5) make it overly complex.


4. τ selection & calibration: How is τ chosen across tasks/methods? Is it per-task tuned on a held-out stream? Show sensitivity of speedup/accuracy to τ and to the judge model.

5. Memory harms: You note degradation at low similarity; can you quantify a retrieval-similarity threshold (e.g., via embedding cosine or PRF) below which memory is suppressed, and show that policy ablation?

6. Generalization beyond synthetic similarity: Can you add a real, temporally ordered workload (e.g., near-duplicate QA logs) to validate speedup under distribution drift?

---

> ### Author Response · Authors · 2025-11-29
>
> Dear Reviewer ueHS,
>
> We appreciate your taking time to review our article and your insightful comments. We want to address your concerns here.
>
>
> **(W1A: incremental significance, expected results).**
>
> **Response:** We respectfully disagree that the contribution is incremental. While it may be intuitive that **identical questions** lead to negligible reasoning cost, our work specifically aims to answer a more nuanced and academically meaningful question:
>
> Does reasoning speedup still occur when questions are **not identical**, and does this depend on the **reasoning paradigm, memory mechanism, question similarity level, or task domain**?
>
> Our framework systematically isolates these variables and provides the first cross-reasoning and memory paradigm, cross-task similarity, cross-task analysis of when reasoning speedup holds or breaks. Thus, our contribution is to resolve when the intuitive “similar -> faster” effect remains valid, and under what conditions it fails.
>
>
> **(W2 & 3, Q1, Q2: Theorem Revision)**
>
> **Response:** We take the reviewer’s suggestions and have substantially **revised both theorems**:
>
> Theorem 1 now explicitly states all necessary assumptions, including the monotonicity of the cost function and the fact that the candidate generation process of method $s$ is fixed across $t$.
>
> Theorem 2 has been rewritten directly under a stochastic-dominance assumption, with independence and DAG-monotonicity listed explicitly in the theorem body.
>
>
> **(W4, Q3) Framework clarity.**
>
> **Response:** We appreciate the reviewer's suggestions. We have expanded the **textual explanations** around Eqs. (3)-(5) and added **explicit intuition** before the formalization (blue text). The illustrations of what cost, score, and adaptive mean in practice are actually already in **Appendix D**.
>
> **(W5 & Q4) Selection of $\tau$**
>
> **Response:** As we mentioned in Appendix G, actually in our experiments, the judge model’s feedback is a categorical value of “surely correct” “may be correct” “surely incorrect”.  τ was set to be “surely correct” in all tasks.
>
> **(W6, Q5) Study task similarity using embedding cosine similarity**
> **Response:** Embedding cosine similarity primarily captures **surface-level or semantic overlap**, but it often fails to identify deeper reasoning-level or conceptual similarity between questions. **This is exactly why our work adopts the S1-S4 similarity taxonomy instead of embedding-based similarity**. Under this formulation, our results validate the necessity of this formulation:
> In 1(identical questions), S2 (Surficially different, semantically same), and S3 (Semantically different, reasoning procedure same) , memory consistently yields efficiency gains; In S4 (reasoning procedure different, underlying knowledge same), efficiency improvements become inconsistent and can even degrade.
>
>
> **(W7 & Q6) Real-World Study.**
> **Response:** Now we included an additional study on a **real-world QA workflow**. We added the results in Appendix I. Specifically, we created an additional test set based on the Stack Overflow database. The Stack Overflow dataset has a label for duplicate questions. We extract a test set of 20 clusters, each cluster with 10 questions marked as duplicated by humans. A strong LLM, GPT-5, is used to evaluate the answer correctness provided with the reference answer (with the highest score on Stack Overflow). Results show that our proposed framework can consistently improve reasoning efficiency in real-world QA streams.
>
> We hope the above clarifications can address your concerns.

---

### Official Review · Reviewer_5AuK · 2025-11-03

**Soundness:** 2
**Presentation:** 1
**Contribution:** 3
**Rating:** 2
**Confidence:** 3

**Summary:**

This paper investigates whether LLMs can reason faster after repeated exposure to similar tasks. The authors formalize this as a reasoning speedup problem, defined by decreasing compute budgets across question sequences with varying similarity. They propose SpeedupLLM, a unified framework integrating adaptive compute budget allocation and memory mechanisms, and provide theoretical guarantees showing non-increasing compute costs with non-decreasing answer quality. Experiments demonstrate that LLMs can achieve measurable reduction in compute cost with memory assistance.

**Strengths:**

* The paper introduces and formalizes the new concept of reasoning speedup by linking compute allocation with question similarity, providing clear operational definitions and measurable evaluation metrics.
* SpeedupLLM elegantly integrates adaptive compute allocation and memory mechanisms, supported by Theorems 1–2.
* The empirical study is comprehensive, which spans four common reasoning tasks (math, code, commonsense, logic), four scaling strategies (Best-of-N, DFS, Self-Refine, Long CoT), and multiple memory types.

**Weaknesses:**

Despite that the paper presents a novel concept of reasoning speedup, the presentation is a main weakness, as stated in the following:
* The motivation (“Can LLMs reason faster through past experience?”) is novel, but the introduction drifts between human analogy, test-time scaling, and memory mechanisms without a clear logical bridge.
* The methodology section is dense with equations and symbol-heavy formulations, but the conceptual flow isn’t intuitive. For example, Section 3.2 mixes adaptive compute allocation and memory mechanisms before giving readers a clear intuition for why these are the right components. The theoretical results (Theorems 1–2) are not well contextualized. Readers must infer the meaning of “non-increasing compute budget” in practical terms.
* Figures (especially Fig. 1–3) are visually complex and lack concise takeaways in captions.

Overall, a clearer structure separating motivation, intuition, and formal results would greatly improve readability and impact. If the authors would improve the presentation, I will consider raising my score. Besides the presentation, there are also some minor weaknesses:
* The question similarity levels are manually designed, potentially oversimplifying real-world semantic overlaps. Using automated metrics (e.g., embedding cosine similarity) would reduce subjectivity.
* The theoretical guarantee assumes non-degrading memory, which may not hold when question similarity is low (e.g., S4 cases).

**Questions:**

Since there is a mainstream using RL to imrove LLM reasoning, can the proposed framework integrate RL to autonomously optimize memory retention and compute budgets over time?

---

> ### Author Response · Authors · 2025-11-29
>
> Dear Reviewer 5AuK,
>
> We appreciate your taking time to review our article and your insightful comments. We want to address your concerns here.
>
> **(W1) logical bridge.**
>
> **Response:** We have substantially **revised the Introduction** to make the logical bridge explicit:
>
> Test-time scaling corresponds to System-2 reasoning, which is allocating more compute to improve accuracy, but at the cost of latency.
> For humans, repeated exposure allows System-2 reasoning to gradually become faster, more “System-1-like”, through memory consolidation.
> This motivates our central question: Can LLMs similarly transition from compute-heavy reasoning to faster, memory-assisted reasoning when encountering related questions?
>
> **(W2) Methodology presentation.**
>
> **Response:** We have now revised the Methodology section to: 1) Introduce the intuitive roles of the two components. 2) Clarify why both components are necessary. 3) Contextualize Theorems 1 and 2 as a decomposition of conditions under which reasoning speedup can occur.
>
> **(W3) Figure captions.**
>
> **Response:** We have revised the figure captions in the updated paper with more concise takeaways.
>
> **(W4) question similarity subjectivity.**
>
> **Response:** Embedding-based metrics (e.g., cosine similarity) often capture only **surface-level** similarity between questions, but they often fail to detect **deeper, reasoning-level equivalence**. Therefore, in this work, we systematically formulate the LLM reasoning task’s similarity into four levels:
>
> S1: (Most similar) Completely same questions (Surficially same).
>
> S2: Surficially different, semantically same questions.
>
> S3: Semantically different questions, reasoning procedure same.
>
> S4: (Least similar) Different reasoning procedure, same underlying knowledge.
>
> The above categorisation of reasoning task similarities is also one key contribution of this paper. We thank the reviewer for giving us this opportunity to make this contribution clearer.
>
> **(W5) The assumption of non-degrading memory may not hold.**
>
> **Response:** We agree with the reviewer that memory does not always help. However, we want to clarify that:
> Our theoretical guarantee is **intentionally conditional**, not universal. The assumption is not an idealization but a characterization of when speedup is achievable.
> This is exactly why we conduct experiments across four similarity levels. Our experiments indeed show that sometimes under low task similarity (S3 or S4), the memory mechanism can slow down the reasoning. In this case, the assumption does not hold, and the conclusion does not hold.
>
> **(Q1)**
>
> **Response:** As a first step toward understanding the speedup behaviour, this work focuses on analyzing its fundamental conditions. Extending SpeedupLLM with RL-based optimization is a promising future direction.
>
> We hope the above clarifications can address your concerns.

---

### Official Review · Reviewer_G8c7 · 2025-11-04

**Soundness:** 2
**Presentation:** 3
**Contribution:** 2
**Rating:** 2
**Confidence:** 3

**Summary:**

This paper explores if LLMs can reason faster via past experience. LLMs need more compute for better reasoning but suffer longer inference time, and existing LLMs lack experience leverage and adaptive compute allocation. It formalizes the problem by defining question similarity (S1-S4) and computing budget, then proposes SPEEDUPLLM, a theoretically guaranteed framework using adaptive compute allocation and memory mechanisms (parametric like SFT, textual like in-context).  Experiments on multiple tasks with 4 test-time scaling methods and 5 memory methods show LLMs can reason faster, with up to 56% compute cost reduction. Higher question similarity and episodic memory boost speedup, and faster reasoning correlates with higher accuracy.

**Strengths:**

- The paper has a clear motivation and is well-written.
- It is novel that the paper analyzes reasoning efficiency from the perspective of similar data.

**Weaknesses:**

- In fact, under the condition of providing similar questions and their reference answers, the increase in the model's reasoning efficiency is reasonable and predictable. If identical questions and their standard answers are provided to the model, LLMs only need to restate the standard answers without any reasoning. Furthermore, we need additional and greater costs to retrieve similar questions, or even to synthesize similar questions and their standard answers. What do you think the practical utility of such evaluation and analysis is for actually improving reasoning efficiency?
- Following up on the previous question, I believe some additional analysis and exploration may be necessary. For example: 1) Similar Q&As are not obtained through synthesis, but retrieved from actual datasets (such as cross-datasets), which is more in line with real-world scenarios (though it seems the effect of improving reasoning efficiency is already weak for S4 Setting). 2) Reference answers are not provided when similar questions are given. After all, similar questions are easy to obtain (e.g., through synthesis), but reference answers are difficult to acquire (synthesis also requires significant costs).

**Questions:**

See Weaknesses

---

> ### Author Response · Authors · 2025-11-29
>
> Dear Reviewer G8c7,
>
> We appreciate your taking time to review our article and your insightful comments. We want to address your concerns here.
>
> **(W1A) Reasoning acceleration under similar questions is predictable:**
>
> **Response**: We agree that in the **extreme case of identical questions with identical answers**, an LLM can trivially restate the solution without reasoning. However, our work specifically aims to answer a more nuanced and academically meaningful question:
>
> Does reasoning speedup still occur when questions are **not identical**, and does this depend on the **reasoning paradigm, memory mechanism, question similarity level, or task domain**?
>
> Our framework systematically isolates these variables and provides the first cross-reasoning and memory paradigm, cross-task similarity, cross-task analysis of when reasoning speedup holds or breaks. Thus, our contribution is to resolve when the intuitive “similar -> faster” effect remains valid, and under what conditions it fails.
>
> **(W1B) Practical utility.**
>
> **Response**: Based on the review, we believe there may be a misunderstanding of our framework. Our method **does not synthesize similar questions**, nor does it require retrieving external Q&A pairs.
> Instead, we model the realistic scenario where an LLM system receives **a stream of user queries**, many of which are near-duplicates or related to past queries.
> In such settings, we only leverage the model’s own past answers, and thus no additional labels, datasets, or synthesized questions are required. Thus, our study evaluates whether the LLM can use its own previous reasoning as memory, and reduce compute on later similar queries.
>
> **(W2A) Evaluation on real datasets.**
>
> **Response**: Now we conducted an additional study on a **real-world QA workflow**. We added the results in **Appendix I**. Specifically, we create an additional test set based on the Stack Overflow database. The Stack Overflow dataset has a label for duplicate questions. We extract a test set of 20 clusters, each cluster with 10 questions marked as duplicated by humans. A strong LLM, GPT-5, is used to evaluate the answer correctness provided with the reference answer (with the highest score on Stack Overflow). Results show that our proposed framework can consistently improve reasoning efficiency in real-world QA streams.
>
> **(W2B) Study when answers are not given to the model.**
>
> **Response**: Similarly to W1B, we want to clarify that reference questions are not an additional dataset; they are from the model’s past queries. The answers are not given to the model, but generated by the model itself.
>
> We hope the above clarifications can address your concerns.

---

### Meta-Review · Area_Chair_MbEE · 2025-12-12

**Summary:**

There are several major concerns raised by the reviewer: 1. The presentation needs to be significantly improved, including organization, logic, explanation, etc. 2. The objective of the theoretical results is not clear. 3. The memory mechanism is not well studied. 4. The technical contributions of some key components of the proposed method are limited.

**Reviewer Concerns:**

Regarding the 4 concerns summarized above, 1. This remains an issue, as the required revisions are too extensive for a standard rebuttal and require re-review. 2 &3 Though the authors explained the objective of the theoretical results and the memory mechanism in the rebuttal, their responses on these points were unconvincing. 4. The authors did not address this concern in the rebuttal.

**Reviewer Scores:**

The rebuttal is insufficient to address the fundamental concerns and thus unlikely to warrant a change in the reviewers' scores.

---

### Decision · Program_Chairs · 2026-01-26

Reject